# Ultrafast all-optical coherent control of single silicon vacancy colour centres in diamond

Jonas Nils Becker[1], Johannes Görlitz[1], Carsten Arend[1], Matthew Markham[2] & Christoph Becher[1]

Complete control of the state of a quantum bit (qubit) is a fundamental requirement for any quantum information processing (QIP) system. In this context, all-optical control techniques offer the advantage of a well-localized and potentially ultrafast manipulation of individual qubits in multi-qubit systems. Recently, the negatively charged silicon vacancy centre (SiV$^-$) in diamond has emerged as a novel promising system for QIP due to its superior spectral properties and advantageous electronic structure, offering an optically accessible $\Lambda$-type level system with large orbital splittings. Here, we report on all-optical resonant as well as Raman-based coherent control of a single SiV$^-$ using ultrafast pulses as short as 1 ps, significantly faster than the centre's phonon-limited ground state coherence time of about 40 ns. These measurements prove the accessibility of a complete set of single-qubit operations relying solely on optical fields and pave the way for high-speed QIP applications using SiV$^-$ centres.

[1] Fachrichtung 7.2 (Experimentalphysik), Universität des Saarlandes, Campus E2.6, 66123 Saarbrücken, Germany. [2] Global Innovation Centre, Element Six Limited, Global Innovation Centre, Fermi Avenue, Harwell Oxford, Didcot, Oxfordshire OX11 0QR, UK. Correspondence and requests for materials should be addressed to C.B. (email: christoph.becher@physik.uni-saarland.de).

To manipulate the internal state of a qubit all-optical coherent control techniques are highly desirable for reasons such as high spatial resolution that allows precise control in multi-qubit systems and high-speed manipulation using ultrafast optical pulses. Ultrafast optical manipulation, on the other hand, implies broadband laser pulses and requires selected electronic systems: While even optical sub-cycle control[1], that is, control over time scales shorter than the qubit frequency splitting, is feasible using a single ultrafast optical pulse spanning, for example, two optical transitions in a $\Lambda$ or $V$-type configuration, attention has to be paid to avoid cross-excitations of unwanted optical transitions due to the large spectral width of the pulse. Therefore these techniques require qubits with isolated electronic levels or large electronic level splittings. For solid state qubits, which are advantageous due to their good scalability, ultrafast optical control has for example, been demonstrated in semiconductor quantum dots[2], as well as the nitrogen vacancy centre (NV) in diamond[3]. However, both systems rely on the application of an additional field to split electronic levels and to realize an optically accessible level scheme. To create an optically addressable $\Lambda$-scheme in quantum dots, charged quantum dots in strong magnetic fields have to be used[4] and typical electron spin coherence times are on the order of one nanosecond[5,6] without employing nuclear field locking techniques or spin echo sequences. Moreover, optical control of the NV relies on the presence of a strain or electric field which lifts the orbital degeneracy in the excited state to create a $V$-type level configuration[3].

As qubits we here consider confined spin impurities in a spin-free diamond host lattice, known as colour centres. They have proven to be promising systems for QIP, as well as sensing applications, with the negatively charged nitrogen vacancy centre (NV$^-$) being the most prominent example[7,8]. Over the past couple of years the SiV$^-$ emerged as a steadily growing competitor to the NV$^-$ due to its superior spectral properties such as intense, narrow zero phonon line emission at ambient and cryogenic temperatures as well as small Huang–Rhys factors down to $S = 0.08$ (refs 9,10). The SiV$^-$ features a unique molecular structure with an interstitial silicon atom in between two unoccupied carbon sites. The resulting inversion symmetry of this split-vacancy configuration is responsible for the high spectral stability of the SiV$^-$ by suppressing first-order Stark shifts and thus allows for the generation of indistinguishable photons from separate SiV$^-$ centres[11]. In previous studies[12,13], the electronic structure of the defect has been investigated in great detail revealing an $S = \frac{1}{2}$ system. Due to a strong spin orbit coupling[12], the centre features twofold orbitally split and spin degenerate ground and excited states at zero magnetic field (Fig. 1a) leading to the characteristic four-line fine structure in the zero phonon line emission spectrum at 737 nm (Fig. 1b). These orbital states can be used to construct an SiV$^-$ based qubit even without the need of an external magnetic field and the large splitting of $\delta_g/2\pi = 48$ GHz in the ground and $\delta_e/2\pi = 259$ GHz in the excited state allows for ultrafast optical coherent control. Previous studies determined the ground state spin coherence time of the centre to be on the order of $T_2^* = 35$–45 ns (refs 14,15). This coherence time is limited by the orbital coherence time $T_1^{\mathrm{Orbit}} = 35$ ns (ref. 15), which results from a single-phonon vibronic process causing transitions between the orbital components of the ground state manifold[16]. Hence, by eliminating these processes in future devices coherence times can be further extended. This can be achieved by either cooling the samples below 1 K or by manipulating the phononic environment of the centre using phononic nanostructures[16]. The rapid development of nanofabrication[17,18] and growth techniques[19] potentially allows for the fabrication of diamond based phononic band gap materials[20] or small nanodiamonds to suppress ground state thermalization. As a result, coherence times in the millisecond range seem feasible as the pure spin relaxation time has been determined to be $T_1^{\mathrm{Spin}} = 2.4$ ms (ref. 15).

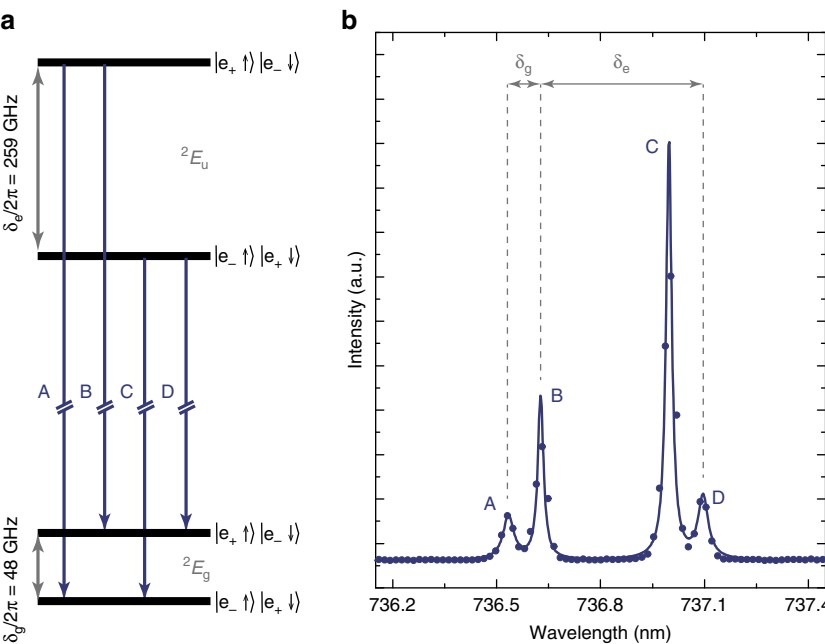

**Figure 1 | Spectrum and Electronic Structure of the SiV$^-$.** (a) Electronic level scheme of the SiV$^-$ at zero magnetic field consisting of an orbitally split but spin degenerate ground and excited state doublet. Large level splittings of 48 GHz in the ground and 259 GHz in the excited state, mainly caused by spin orbit coupling, are characteristic for the SiV$^-$ and render the system ideal for manipulation using broadband laser pulses. (b) Photoluminescence spectrum of the four zero phonon line transitions of the SiV$^-$ centre used throughout this study. The spectrum is obtained under non-resonant excitation at $\lambda_{ex} = 690$ nm and at 4 K. The values of the spectral splitting between transitions A and B as well as A and C increase for higher crystal strain and, in this case, indicate an unstrained centre.

Here we present ultrafast and all-optical resonant as well as Raman-based coherent control of a single $SiV^-$ using laser pulses with lenghts of 1–30 ps. This demonstrates the feasibility of hundreds to thousands of gate operations even in the phonon-limited regime with coherence times of several tens of nanoseconds, paving the way for high-speed QIP applications using $SiV^-$ centres.

## Results

**Resonant coherent control.** A universal single-qubit gate requires control of the angle (U(1) control) as well as the axis of rotation (SU(2) control) of a qubit state. First, we demonstrate resonant angular control by coherently driving Rabi oscillations between the ground and the excited state of the centre using 12 ps laser pulses. Figures 2a,b show the fluorescence signal from the excited state for variable pulse areas of the picosecond laser resonant with transitions C and B (Fig. 1). High contrast Rabi oscillations are evident in the photon count rate traces of both transitions with visibilities exceeding 90% and are well-fitted by a four level density matrix model (solid lines, Supplementary Figs 1–3, Supplementary Note 1). Coherent rotations up to $\Theta \approx 10\pi$ for transition C and up to $\Theta \approx 6\pi$ for transition B are observed without any significant damping, with the rotation angles being limited by the available laser power. Residual background fluorescence of the diamond causes slight upwards slopes in both curves. As a direct measurement of the absolute excited state population of the SiV is not feasible, we estimate the $\pi$-pulse fidelity from the theoretical model. Our simulation indicates that fidelities on the order of 95% are reached in this measurement, limited by a slight cross-excitation of neighbouring optical transitions. This can be further improved in future experiments by using slightly longer and thus narrower resonant rotation pulses.

The oscillations shown in Fig. 2 demonstrate rotation about the $x$-axis of the Bloch sphere. However, full SU(2) control requires rotation around a second axis. This can be achieved by exploiting the free precession of the state about the vertical $z$-axis of the Bloch sphere by employing a Ramsey-type pulse sequence consisting of two $\frac{\pi}{2}$-pulses separated by a variable delay time. As the experimental resonance frequency is about 406.8 THz, the oscillations in the signal are extremely fast and no single Ramsey fringes have been resolved. However, by fine-tuning the temporal spacing of the two pulses for a number of fixed delays it is possible to determine the maximum and minimum count rate at each fixed delay point and therefore to measure the upper and lower envelopes of the Ramsey curve which are shown in Fig. 3a for transitions B and C. Simulations (solid lines) using the four-level model show that two effects are responsible for the decrease of the Ramsey fringe amplitudes in Fig. 3b. First, spontaneous decay and thermalization effects lead to a decoherence of the system. Secondly, a pure dephasing of the excited states of about $\gamma/2\pi = 160$ MHz, most likely caused by residual phonon broadening, has to be added to fit the observed decrease in fringe amplitude. Calculations of the excited state coherence times from these amplitudes yield $T_2^* = 1,044$ ps for the lower and $T_2^* = 398$ ps for the upper excited state, with the latter one decohering faster due to a rapid decay into the lower state. The resulting linewidths $\Delta v = (\pi T_2^*)^{-1}$ of 305 MHz for C and 799 MHz for B are in reasonable agreement with the linewidths measured by Arend *et al.* using photoluminescence excitation spectroscopy[21]. This indicates that coherently driving the $SiV^-$, even with intense pulsed fields, does not alter its coherence properties, e.g., due to the generation of charges in the environment of the centre, as it is a common problem for quantum dots. Moreover, no decrease in fluorescence count rates

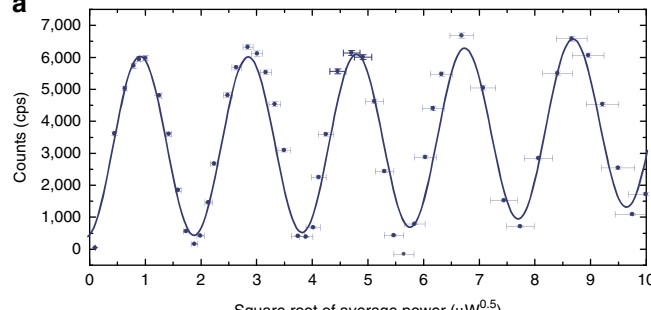

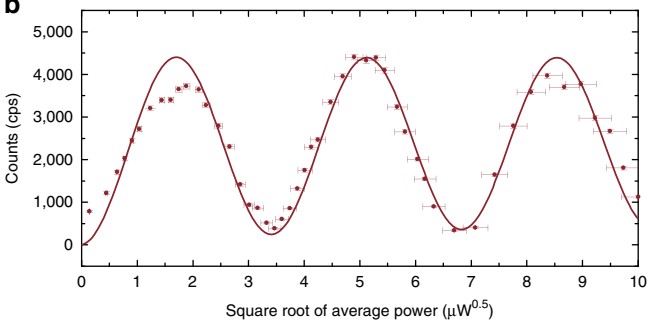

**Figure 2 | Optical one-photon Rabi oscillations.** Measured photon counts as a function of average laser power (after subtraction of background fluorescence mainly caused by milling of solid immersion lenses) for the picosecond laser pulses resonant with (**a**) transition C and (**b**) transition B. In both cases pulses with 12 ps length ($\Delta v = 20$ GHz) and a double-sided exponential temporal shape (due to filtering with a Fabry–Perot etalon) have been applied (x-errors (s.d.): measured relative laser power stability, y-errors (s.d.): Poisson-distributed count rate errors $\sqrt{y}$). The data in both graphs is modelled by a four-level density matrix model (solid lines) including spontaneous decays between the excited and ground states as well as the phonon-induced decay and thermalization processes within the ground and excited state manifolds. All rates have been measured experimentally and no additional free parameters have been employed to model the Rabi oscillations (Supplementary Figs 1–3, Supplementary Note 1).

for high rotation angles is observed in the Rabi oscillations in Fig. 2, indicating no power-dependent ionization of the centre (for example, into the $SiV^0$ charge state) as it is observed for NV centres[3].

**Raman-based ground state control.** The resonant control techniques presented so far require excitation of the $SiV^-$ and hence coherence times of such a qubit are ultimately limited by the excited state dynamics. To harness the longer ground state coherence time of the $SiV^-$, coherent rotations solely within the ground state manifold are desirable. This can be achieved by driving a $\Lambda$-type level scheme between the two ground and a common excited state. A Raman process then leads to a coherent transfer of population between the ground states. To minimize an unwanted population of the excited state, both arms of the $\Lambda$-scheme are far-detuned from one-photon resonance, while still fulfilling a two-photon resonance condition. We here simultaneously couple both transitions of the $\Lambda$-system employing a single broadband laser pulse with a length of only 1 ps, red-detuned by $\Delta = 500$ GHz from the lower excited state[2,22]. Furthermore, initialization of the population into the lower ground state and readout is achieved by optical pumping of transition D using resonant 200 ns long laser pulses

 

(Supplementary Fig. 4; Supplementary Note 2). After pumping, the population in the upper ground state reaches a minimum value of 22%. The spectral arrangement of the two lasers relative to the SiV$^-$ spectrum is depicted in Fig. 4a and the pulse sequence, as well as two count rate traces at two different Raman pulse powers are shown in Fig. 4b. The two branches of the $\Lambda$-system have different transition dipole moments and polarizations. In an unstrained SiV$^-$ such as the one investigated in this work, the inner lines B and C are linearly polarized. The outer lines A and D are circularly polarized, however, due to the projection of their $xy$-dipole onto the sample surface also appear linear (with a reduced contrast) and orthogonal to line B and C (ref. 12). To maximize the transfer efficiency the driving strengths of both Raman transitions have to be equalized. This is carried out by rotating the polarization of the Raman beam at a fixed pulse power. At the optimized polarization, the population of the upper ground state is measured for varying Raman pulse areas. The resulting curve is depicted in Fig. 4c, showing a two-photon Rabi oscillation with rotation angles of up to $\Theta \approx 2\pi$, limited by the rotation laser power. As before, the data is fitted well by our model (solid curve) with the same set of experimentally measured parameters that have been used for the one-photon experiments and with the relative driving strength of the two Raman transitions being the only free parameter. The model also includes transfer via the

second $\Lambda$-scheme between the ground states and the upper excited state, from which the laser is detuned by $\Delta + \delta_e = 759$ GHz. Simulations indicate that the decrease in visibility for large angles is not due to ground state decoherence but due to slight resonant excitation of the excited states for large pulse areas (Supplementary Fig. 5; Supplementary Note 3). In future experiments this can be minimized by modifying the laser system to provide longer pulses with a duration of a few ps. To demonstrate full SU(2) control of the ground state, also a Ramsey-type experiment has been performed and the resulting data are shown in Fig. 4d. Oscillations at the frequency of the

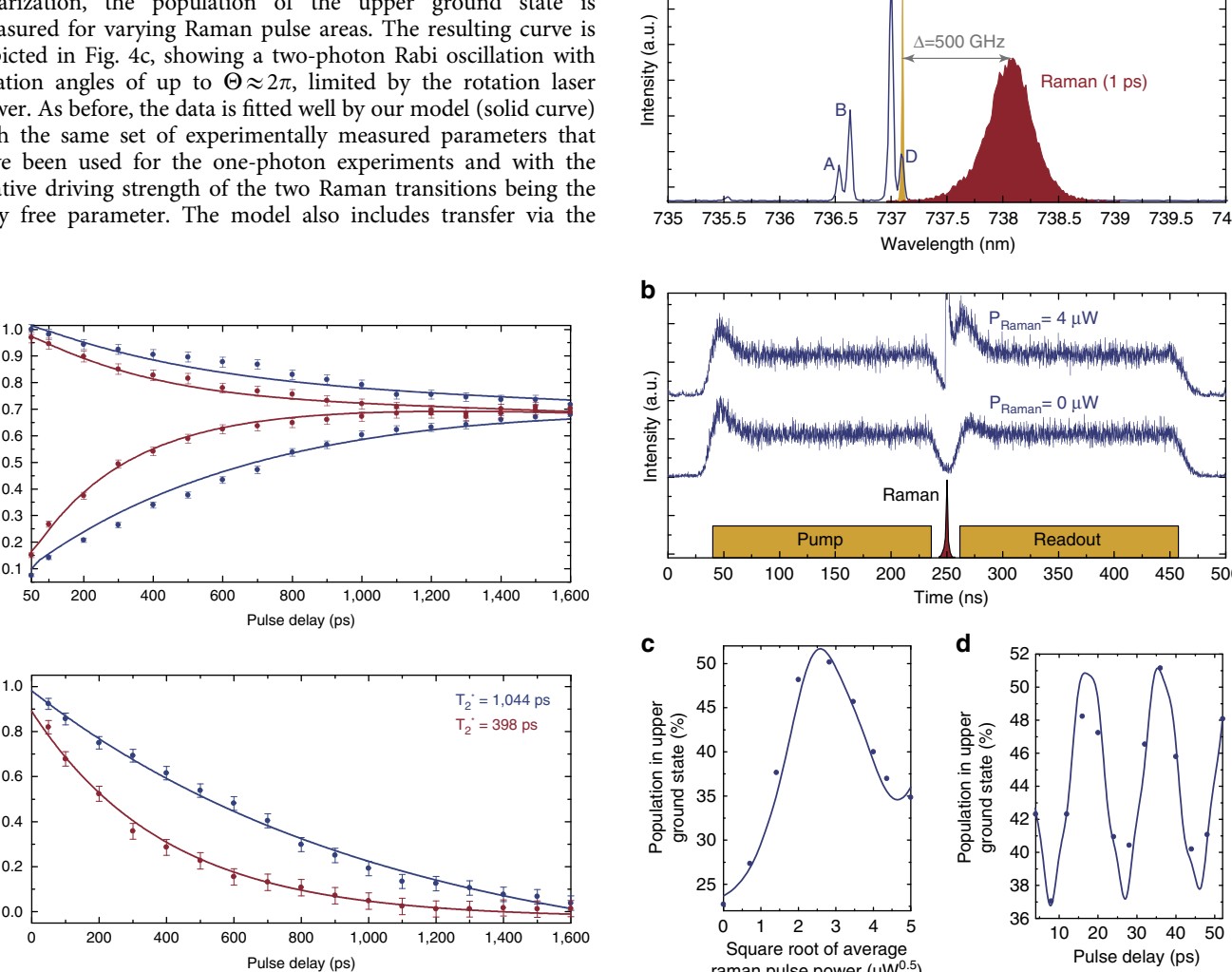

**Figure 3 | Optical one-photon Ramsey interference.** Interference fringes can be observed by applying a sequence of two picosecond $\frac{\pi}{2}$-pulses with variable delay, a so-called Ramsey-sequence. (**a**) Upper and lower envelopes of the observed interference pattern for transition C (blue) and transition B (red). Individual fringes are not resolved due to their short oscillation period at optical frequencies. (**b**) The excited state coherence times can be extracted from the decay of the interference fringe amplitude. From the amplitude decay on transition C (blue) a coherence time of $T_2^* = 1,044$ ps for the lower excited state can be extracted by simulating the decay using the four-level density matrix model (solid lines). Analogously on transition B (red), a coherence time of $T_2^* = 398$ ps is measured for the upper excited state ($y$-errors (s.d.): Poisson-distributed count rate errors $\sqrt{y}$).

**Figure 4 | Raman-based population transfer between ground states.** (**a**) Spectrum displaying spectral position of the SiV$^-$, as well as width and detuning of the CW Pump/Readout and pulsed Raman rotation laser involved. (**b**) Schematic pulse sequence and measured fluorescence response for the Raman beam switched off as well as $P_{Raman} = 4\,\mu$W (red). (**c**) Transferred population in upper ground state as a function of the average power in the Raman beam. (**d**) Ramsey interference generated by two subsequent Raman pulses verifying coherent transfer. Solid lines show simulations using the four-level density matrix model with the relative driving strength of both Raman transitions as the only free parameter. The model indicates that the ratio of the driving strengths of transitions C and D in the Raman beam is about 1:0.7 after optimizing the polarization.

ground state splitting of $\delta_g/2\pi = 48$ GHz are clearly visible and are reproduced well by the model (solid line), without any additional free parameters. The slight distortion of the curve is again due to resonant background fluorescence and the population does not reach the optically pumped minimum as a small amount of incoherent transfer occurs[22]. The two-photon $\pi$-pulse fidelity in this measurement is 30%. This fidelity is limited by two factors, the efficiency of the optical pumping sequence as well as width, detuning and polarization of the Raman pulse. After switching off the acousto-optic modulator which generates the pump pulse, a delay of 15 ns was necessary between pump and Raman pulse to avoid overlaps, due to the acousto-optic modulator rise- and fall time. During this time the population in the upper ground state increases from an initial value of about 10.5–22% due to ground state thermalization. Therefore, the initialization fidelity can be improved in future experiments by either reducing the thermalization rate within the ground state manifold or by using a faster modulator which allows shorter delays between the pump and the rotation pulse (Supplementary Fig. 4; Supplementary Note 2). Moreover, optimization of pulse parameters such as width, detuning, polarization and chirp could further improve transfer fidelities of this simple sub-cycle control scheme. Furthermore, in future experiments more complex transfer schemes with two separate rotation fields can be employed, that avoid resonant excitation and allow for more accurate optimization of the relative transition strengths. Moreover, a time delay between both fields can be used to optimize the adiabaticity of the process[23].

## Discussion

In conclusion we demonstrated resonant and off-resonant Raman-based coherent control of the rotation angle and axis of the orbital degree of freedom of an SiV$^-$ employing ultrafast rotation pulses as short as 1 ps. All the experiments in this work have been carried out on the orbital states of the SiV$^-$ without the need of a strong magnetic field. However, we'd like to emphasize that the techniques presented here can directly be applied to the spin states of the SiV$^-$. Although the coherence time is reduced for intermediate field strengths, we demonstrated that it recovers in the high field regime above approximately 5 T (ref. 14). This regime is required in any case to achieve large Zeeman splittings to enable the use of ultrashort pulses. The results presented here provide the basis for a number of exciting QIP applications with the SiV$^-$ such as cavity-assisted Raman transfer schemes[24,25], coherent spin-photon interfaces[26], optical quantum memories[27,28] and quantum gates based on geometrical phase acquisition[29].

## Methods

**Sample properties.** The sample consists of a type IIa high-pressure-high-temperature diamond with a cleaved (111) main surface. The sample is implanted with a dose of $10^9$ ions per cm$^2$ of $^{28}$Si$^+$ ions at an energy of 900 keV, resulting in an $\sim$50 nm thick layer of SiV$^-$ centres 500 nm below the diamond surface. After implantation the sample is annealed for 3 h at 1,000 °C in vacuum and for 1 h at 465 °C in air and cleaned in peroxomonosulfuric acid to remove graphite residues. To further enhance the collection efficiency from the sample, solid immersion lenses with a diameter of 1 μm are milled into the diamond surface using focused ion beam milling. To cure the host lattice damage induced by the focused ion beam milling process the sample is annealed a second time using the above-mentioned protocol. The resulting colour centre density is $\sim$0.2–0.4 SiV$^-$ per μm$^2$ and the inhomogeneous distribution of the centres is on the order of $\Delta\nu = 1$ nm, mainly limited by strain induced during implantation and FIB milling. For this work an unstrained SiV$^-$ has been selected from confocal microscopic scans and spectroscopic investigations. After identifying a suitable centre, its ground state coherence time is probed by measuring the orbital ground state relaxation rates using an optical pumping sequence. For the SiV$^-$ investigated throughout this work, this orbital coherence time has been measured to be $T_1^{Orbit} = 34.3 \pm 5.6$ ns (Supplementary Fig. 3; Supplementary Note 1).

**Optical setup.** The sample is investigated in a homebuilt confocal microscope equipped with a liquid helium flow cryostat and an NA = 0.9 objective. All investigations have been carried out at a temperature of 5 K. For the ultrafast control experiments a tunable mode-locked Ti:sapphire laser provides hyperbolic secant pulses with a length of 1 ps at a repetition rate of 80 MHz. The ground state Raman transfer is carried out without further filtering at a wavelength of 738 nm for the Raman pulse. For the resonant picosecond control experiments the bandwidth of these pulses has to be reduced to avoid excitation of multiple transitions. Therefore the pulses are filtered using a monolithic Fabry–Perot etalon (F = 50, FSR = 1,000 GHz, $\Delta\nu = 20$ GHz). This results in Lorentzian-shaped pulses with a length of 12 ps suitable to selectively excite the fine structure transitions of the SiV$^-$. Pulse intensities are stabilized using a feedback loop formed by a combination of a voice coil steering mirror in front of a single mode optical fibre and a photodiode after the fibre connected to a PID controller. For the Raman transfer experiments an initialization of the population into the lowest ground state, as well as readout of the population in the upper ground state after the transfer is necessary. This is achieved by excitation of transition D using a resonant continuous wave extended cavity diode laser stabilized to a wavemeter. Initialization and readout pulses are generated from this laser using an acousto-optic intensity modulator in single-pass configuration. To generate a sufficiently long time interval for initialization and readout the repetition rate of the mode-locked laser has been reduced to 1 MHz using an electro-optic pulse picker cell consisting of a Pockels cell in between two Glan-Taylor prisms providing a pulse suppression greater than 1,000 when the cell is switched off. State detection is carried out by detecting the light emitted from the SiV$^-$ phonon sideband using a single photon detector attached to a fast counting electronics. The trigger signal provided by the pulsed laser or the pulse picker cell, respectively, is used to synchronize the picosecond laser pulses with the pump and readout pulses from the extended cavity diode laser, as well as with the counting electronics. The data in the one-photon experiments is acquired by directly measuring the photon count rate averaged over 500 ms per data point. In the two-photon experiments, time-resolved fluorescence count rate traces for the entire sequence duration of 1 μs have been recorded with a total integration time of 200 s per trace. The population of the upper ground state then is extracted from the rising edge of the fluorescence signal of the readout pulse.

**Data availability.** All relevant data are available from the corresponding author on request.

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

## Acknowledgements

This research has been partially funded by the European Community's Seventh Framework Programme (FP7/2007-2013) under Grant agreement no. 611143 (DIADEMS). Ion implantation was performed at and supported by RUBION, the central unit of the Ruhr-Universität Bochum. We thank D. Rogalla for the implantation, C. Pauly for the fabrication of solid immersion lenses as well as M. Bock for technical assistance with the picosecond laser system. Moreover, we thank M. Atatüre, B. Pingault, G. Morigi, J. Eschner and E. Neu for helpful discussions throughout all stages of this work.

## Author contributions

J.N.B. designed, built and conducted the experiment with help from C.A. Theoretical simulations and analysis of experimental results have been carried out by J.N.B and J.G. The sample has been provided by M.M. The manuscript has been written by J.N.B., J.G. and C.B. C.B. conceived the experiment. All authors discussed the results and commented on the manuscript.

## Additional information

**Competing financial interests:** The authors declare no competing financial interests.

**Publisher's note**: 

