## [Peer Review File · Nature Communications]

Editorial Note: this manuscript has been previously reviewed at another journal that is not operating a transparent peer review scheme. This document only contains reviewer comments and rebuttal letters for versions considered at Nature Communications. Mentions of prior referee reports have been redacted.

Reviewers' comments:

Reviewer #1 (Remarks to the Author):

The authors have done a commendable job of responding to my comments in [redacted]. I now enthusiastically recommend publication in Nature Communications. I encourage the editors to highlight this technically masterful work in a suitable manner.

Reviewer #3 (Remarks to the Author):

The revised manuscript transferred to Nature communications by Becker et al. addressed most of my concerns, while there is still one about the detuning of the two photon Raman process. If my understanding is correct, the contribution of the second pathway is about $1/(759/500)^2 \sim 0.43$, which is not a negligible fraction. Therefore I can not recommend for publication before the authors adress this issue.

Reviewer #3 (Remarks to the Author):

The revised manuscript transferred to Nature communications by Becker et al. addressed most of my concerns, while there is still one about the detuning of the two photon Raman process. If my understanding is correct, the contribution of the second pathway is about $1/(759/500)^2 \sim 0.43$, which is not a negligible fraction. Therefore I can not recommend for publication before the authors adress this issue.

We thank the Reviewer for his thorough work. We reinvestigated the role of the second Lambda-scheme giving rise to the second pathway in greater detail (and beyond simple scaling arguments) to further clarify our previous statement on that matter. It turns out that the contribution of the second Lambda-scheme is greater than initially assumed, but still is not significant: The transfer rate by the second pathway is considerably smaller as compared to the primary Lambda scheme due to a significant difference in transition dipole moments of the involved transitions and the difference in detunings. Beyond the

09.09.2016 | Seite 2

rather simple picture of transition rates we further explored coherent multi-level effects, i.e. the possibility of destructive path interference leading to a reduction of the effective Raman-Rabi-frequency. Nevertheless, such interference effects should not occur in the experiments presented in this manuscript as both arms of the Lambda-systems are driven with a single laser pulse and, furthermore, the time scales are very short.

To further explore the potential occurrence of such effects we performed simulations of Raman-Rabi-oscillations for a single and a double Lambda-system, assuming parameters as in our experiments. From these simulations we found a non-negligible contribution of the second excited state to the decoherence of the system. As stated in our initial manuscript, we reconfirmed that this is not due to reduced adiabaticity of the transfer but due to additional resonant excitation caused by an overlap of the leading edge of the detuned rotation pulse with the zero-phonon-line transitions. Although the increased decoherence reduces the attainable Raman transfer fidelities in the current experiment, this is not a fundamental issue: by employing narrower rotation pulses, this decoherence source can be strongly suppressed.

To clarify this matter in the manuscript we included an additional Supplementary Figure 5 displaying the simulation of Raman-Rabi-oscillations for a single- and a double-Lambda system. Moreover, we added Supplemental Note 3 discussing the differences in Raman-Rabi-frequencies, possible path interference and additional resonant excitation of the system due to the broad rotation pulse.

Furthermore, to comply with the formatting guidelines of Nature Communications we made the following changes to the manuscript:

- we removed the punctuation from the title. The new title now reads *"Ultrafast all-optical coherent control of single silicon vacancy colour centres in diamond"*
- we added a small paragraph summarizing results and conclusion of the paper to the end of the "Introduction" section: *"We here present ultrafast and all-optical resonant as well as Raman-based coherent control of a single SiV⁻ using laser pulses with lengths of 1-30ps. This demonstrates the feasibility of hundreds to thousands of gate operations even in the phonon-limited regime with coherence times of several tens of nanoseconds, paving the way for high-speed QIP applications using SiV⁻ centres."*
- we introduced subheadings in the "Results" section
- we are now referencing to specific elements of the supplementary information
- we introduced subheadings to the methods section and added a data-availability-statement

09.09.2016 | Seite 3

- we changed the citation style to comply with the style of Nat. Commun.
- we defined error bars in the legends of figure 2 and 3 as s.d.
- we replaced the preprint of U. Jantzen et al. in the references of the main text by the final publication
- we rearranged the elements in the supplemental material to comply with the style guidelines
- supplemental material: we added the sentences "In Supplementary Fig4(b) the pump pulse contrast (blue) as well as the population in the upper ground state (red) is displayed for increasing pump Rabi frequencies. While the population rapidly decreases, the contrast of the pump pulse leading edge peak evolves more slowly and levels off at about 40% of the peak count rate." to refer to Supplementary Fig.4(b) in the text.
- we added supplementary figure 5 and supplementary note 3 to address the remark of Reviewer #3

REVIEWERS' COMMENTS:

Reviewer #3 (Remarks to the Author):

The authors have addressed my issue and I recommend it for publication on Nature Communications.